# Biomedical Data Commons (BMDC) prioritizes B-lymphocyte non-coding genetic variants in Type 1 Diabetes

**Samantha N. Piekos**[1,2]*, **Sadhana Gaddam**[1], **Pranav Bhardwaj**[3], **Prashanth Radhakrishnan**[2], **Ramanathan V. Guha**[2], **Anthony E. Oro**[1]*

**1** Program in Epithelial Biology, Stanford University, Stanford, California, United States of America, **2** Google Data Commons, Mountain View, California, United States of America, **3** Department of Statistics, Stanford University, Stanford, California, United States of America

* Samantha.Piekos@isbscience.org (SNP); oro@stanford.edu (AEO)

**Data Availability Statement:** Raw data for the GM12878 cohesin HiChIP, and ATAC-seq datasets were obtained from the Gene Expression Omnibus (GEO): GSE80820.[59] The GM12878 ATAC-seq raw dataset was taken from repository GEO:

## Abstract

The repurposing of biomedical data is inhibited by its fragmented and multi-formatted nature that requires redundant investment of time and resources by data scientists. This is particularly true for Type 1 Diabetes (T1D), one of the most intensely studied common childhood diseases. Intense investigation of the contribution of pancreatic β-islet and T-lymphocytes in T1D has been made. However, genetic contributions from B-lymphocytes, which are known to play a role in a subset of T1D patients, remain relatively understudied. We have addressed this issue through the creation of Biomedical Data Commons (BMDC), a knowledge graph that integrates data from multiple sources into a single queryable format. This increases the speed of analysis by multiple orders of magnitude. We develop a pipeline using B-lymphocyte multi-dimensional epigenome and connectome data and deploy BMDC to assess genetic variants in the context of Type 1 Diabetes (T1D). Pipeline-identified variants are primarily common, non-coding, poorly conserved, and are of unknown clinical significance. While variants and their chromatin connectivity are cell-type specific, they are associated with well-studied disease genes in T-lymphocytes. Candidates include established variants in the HLA-DQB1 and HLA-DRB1 and IL2RA loci that have previously been demonstrated to protect against T1D in humans and mice providing validation for this method. Others are included in the well-established T1D GRS2 genetic risk scoring method. More intriguingly, other prioritized variants are completely novel and form the basis for future mechanistic and clinical validation studies The BMDC community-based platform can be expanded and repurposed to increase the accessibility, reproducibility, and productivity of biomedical information for diverse applications including the prioritization of cell type-specific disease alleles from complex phenotypes.

## Author summary

The fragmentation of datasets prevents repurposing due to time-intensive data cleaning and joining. This is especially true for Type 1 Diabetes for which the genetic contributions

GSE47753.[56] The GM12878 H3K27ac HiChIP as well as the 1° Naïve T, Th17, and Treg cell ATAC-seq and H3k27ac HiChIP datasets were downloaded from GEO: GSE101498.[57] The GM12878 transcription factor ChIP-seq datasets were generated as part of the ENCODE consortium and the bam files can be obtained from GEO (accession numbers detailed in S5 Table).[58] The GM12878 RNA-seq dataset was downloaded from GEO: GSM591661. The github repository datacommonsorg (https://github.com/datacommonsorg) contains the code for the Data Commons schema, python API, and the import of data into Data Commons, including the data which was used in subsequent analyses this paper. Code for running the SNP pipeline and downstream analyses including analyses using Biomedical Data Commons is available under the github repository Oro_Lab_Stanford/SNP_Prioritization_Pipeline (https://github.com/OroLabStanford/SNP_Prioritization_Pipeline)."

**Funding:** This work was funded by NIH R01 ARO73170 (A.E.O.; https://www.niams.nih.gov/), the Bio-X Stanford Interdisciplinary Graduate Fellowship awarded through the Stanford University School of Medicine (S.N.P.; https://biox.stanford.edu/research/phd-fellows), and Google Data Commons (R.V.G.; https://datacommons.org/). The funders had no role in study design, data collection and analysis, decision to publish, or preparation of the manuscript.

**Competing interests:** The authors have declared that no competing interests exist.

from B-lymphocytes, a specific type of white blood cells, remain understudied. Here, we create Biomedical Data Commons (BMDC), a knowledge graph, which maps datasets to common entities making them easy to search using queries. We also built a genetic variant prioritization pipeline that uses multi-dimensional 'omics data including three-dimensional connectome data. Using B-lymphocyte cell-type specific data as input, we prioritized variants associated with Type 1 Diabetes. The candidate variants identified are primarily of unknown clinical significance and in the non-coding genome. They are also connected with genes previously implicated in Type 1 Diabetes, suggesting that they affect cell type-specific gene regulation. Some variants in the HLA and IL2RA locus, which are important genomic regions for regulation of immune function, have previously been validated in humans and mice. Other variants have been included in a well-established Type 1 Diabetes genetic risk scoring method. This validates our approach and highlights the novel variants identified that should be prioritized for future clinical and experimental validation. BMDC is a community-based platform that increases the accessibility, reproducibility, and productivity of biomedical information for diverse applications, and our approach is widely applicable for prioritizing variants from other complex diseases.

## Introduction

The explosion over the past decade of high-throughput biomedical genomics data and the universal transition to electronic medical records promises unparalleled disease insights.[1–3] However, this promise has been hampered by problems in data-sharing and integration limiting the productivity and impact any individual biomedical dataset can generate. Currently, no publicly available, queryable central biomedical database exists, highlighting the inefficiency of the current data structure that forces investigators to spend ~80% of their time individually downloading and cleaning data.[4,5].

In addition, big data can have limitations in study design that impact interpretation. For example, genome-wide association studies (GWAS) seek to understand disease pathogenesis by correlating human sequence variation with disease phenotypes,[6,7] however, multiple hypothesis testing, linkage disequilibrium, and limited or heterogeneous disease populations limit the resolution of these studies. These issues lead to an overemphasis on variations with relatively rare minor allele frequencies in diseases. GWAS studies also fail to address the issues of polymorphism or polygenetic nature of complex diseases. Moreover, there are limited tools to assess the relative importance of the non-coding genome despite it composing 98.5% of the genome. Prioritization methods have traditionally used evolutionary conservation, which focuses on the 1.5% of the genome consisting of highly conserved protein coding regions.[8,9] Sequence conservation is ill-equipped to assess non-coding sequences, which are under increased evolutionary pressure.[10] and fails to account for the affected set of gene targets given the non-linear three-dimensional nature of the genome. Merging multidimensional omics data with disease sequence variation would allow improved functional insights into associative data.

To address these problems, we collaborated with Google Data Commons to develop Biomedical Data Commons (BMDC), a queryable biomedical knowledge graph that integrates publicly available biomedical data. This solves the problem of data cleaning and integration across multiple datasets and domains. It also allows users to easily explore and analyze the data using BMDC's application programming interface (API). Here, we apply BMDC to quickly provide publicly available information on candidate Type 1 Diabetes (T1D) genetic variants

that were prioritized using a pipeline that uses private multi-dimensional cell type-specific data as input. The data extracted using BMDC aiding in data interpretation of candidate T1D variants.

T1D is a common chronic autoimmune disease with an incidence of 25 per 100,000 in Europe and the US. The underlying pathogenesis of T1D involves tolerance-breaking and immune-mediated destruction of pancreatic β-cells resulting in poor blood glucose homeostasis.[11] Genetic and environmental studies indicate considerable genetic heterogeneity, leading to the development of subsets or endotypes to better classify patient risk.[12–14] More recently a genetic risk score (GRS2) of 67 single nucleotide polymorphisms (SNPs) predicted 77% of those who would develop T1D, with increased ability to determine early-onset diabetes.[15] This reinforced the genetic basis of T1D and highlighted the need for better insights into pathogenesis.

Intense investigation into the T1D pathogenesis has identified central roles for islet β-cells and T-lymphocyte subsets in establishing and maintaining the autoimmune state.[11,14] The majority of all research on T1D focuses on T-lymphocytes despite contributions from other cell types known to play a role in T1D, including B-lymphocytes[14,16]. These additional cell types have been largely overlooked and their precise role and pathogenesis remain perplexing. This is especially true for B-lymphocytes, which appears to play a role in a subset of T1D patients.

Patients with agammaglobulinemia or hereditary B-lymphocyte deficiency can still develop T1D [7,17] and T1D-associated antibodies are not pathogenic, arguing against humoral immunity. By contrast, B-lymphocytes are essential for diabetes development in non-obese diabetic mice.[18] Moreover, children developing the condition before the age of 7 years demonstrate a CD20[hi] B-lymphocyte infiltration phenotype (CD20[hi] phenotype; T1DE1)[19,20], implicating B-lymphocytes in disease acceleration. Consistent with this role, anti-CD20 treatment in mice reverses disease onset,[21] and anti-CD20/anti-CD3 prevents disease,[22] while 1-month anti-CD20 treatment in humans delays ß-cell loss as measured at 1 year.[23] These studies support a role for B-lymphocyte-dependent antigen presentation and T-activation that contributes to an early disease onset, but the mechanistic insights into B-lymphocyte dysfunction remains poorly understood. We therefore focused on B-lymphocytes rather than the well-studied T-lymphocytes when we applied BMDC to investigate the role of genetics in B-lymphocyte dysfunction in T1D.

To accomplish this, we developed a pipeline to prioritize genetic variants that leverages the hypothesis that enhancers are cis-acting regulatory regions that modulate gene expression. The enhancer landscape is cell type-specific, therefore a genetic variant that modifies an enhancer's function is a potential mechanism by which genetic variants only impact gene expression in a subset of cells. This has previously been demonstrated in the context of Alzheimer's Disease.[24] Deletion of a non-coding enhancer containing Alzheimer's Disease variants impacted expression of a distal target gene in microglia, but not neurons or astrocytes. This demonstrates how a genetic variant can act in a cell type-specific manner. We apply the same concept here in the context of T1D and expect to find genetic variants in non-coding enhancers that are cell type specific. This approach can provide insight into the role of genetic variants in gene misregulation of specific cell types like B-lymphocytes distinct from other involved cell types including T-lymphocytes.

First, we demonstrate that BMDC can be leveraged to address these myriad analytical shortfalls of big data. Then, we expand upon previous efforts using three-dimensional chromatin data to understand the role of the non-coding genome.[25] Our approach is prioritizing non-coding genetic variants based on the biology of a specific cell type of interest in contrast to the current standard of evolutionary conservation. It also assigns non-coding variants to both a

regulatory element and their distal gene targets based on the unique three-dimensional chromatin conformation of that cell type. This provides a potential mechanism of action for how a genetic variant impacts gene expression. This pipeline is deployed to prioritize T1D-associated genetic variants in B-lymphocytes. Then we interrogate the publicly available knowledge on these variants using BMDC. We confirm previously identified regions associated with early onset T1D associated with high B-lymphocyte infiltration. We also demonstrate their association with functional transcriptional regulation suggesting an underlying activation state of B-lymphocytes in T1D pathogenesis. This approach can be applied to other phenotypes, and the BMDC community-based platform can be expanded upon leading to an increase in the accessibility, reproducibility, and productivity of biomedical data.

## Results

### Biomedical Data Commons integrates multiple data types into a single graph

Genomic, epigenomic, and transcriptomic data from eight databases have been integrated into a queryable knowledge graph (Figs 1A and 1B and S1, and S1 and S2 Tables). Raw data was converted into Meta Content Format (http://www.guha.com/mcf/mcf_spec.html)—mapping to ~50.7 billion unique entities and ~50.0 billion triples (node-edge-node)—and ingested into the Biomedical Data Commons. Existing schema from schema.org (https://schema.org/), which powers ~40% of all websites, was expanded to accommodate this biomedical data [26]. This data can be accessed using one of Google Data Commons APIs (https://docs.datacommons.org/api/), which are powered by BigQuery;[27] https://cloud.google.com/bigquery). The graph is accessible via the Data Commons browser (https://datacommons.org/) and documentation can be found on the browser (https://docs.datacommons.org/) and github (https://github.com/datacommonsorg).

A major advantage of the Biomedical Data Commons is that it integrates data into a searchable format that can then be used to extract information on a subset of the graph at scale. For example, using the python API and SPARQL we were able to search a subset of the graph that we were interested in—Gene, GeneticVariant, and GeneGeneticVariantAssociation nodes and edges (Fig 1C, top panel)—and extract only Gene and Genetic Variants that were associated with each other in whole blood by GTEx (Fig 1C, bottom panel). Extracting data using the Data Commons python API is faster by one to four orders of magnitude depending on the query and requires code of lower cyclomatic complexity than the standard data scientist approach of cleaning and preparing the raw data to answer basic biomedical questions (Figs 1D and S2A).[28]

### SNP Prioritization Pipeline integrates multidimensional 'omics data to form gene regulatory networks and prioritize biological validation targets

We developed a pipeline that takes multiple data types—regulatory elements, genetic variants, genes, and three-dimensional chromatin structure—to build gene regulatory networks and prioritize genetic variants and genes for biological validation experiments (Fig 2). This pipeline is deployed using input genetic variants from a disease of interest and epigenomic data from a cell type of interest. The regulatory elements that contain genetic variants that are in close physical proximity to a linearly distal gene target based on the three-dimensional chromatin structure are associated with one another to form regulatory elements—genetic variant—gene trios following the first step of the pipeline. These are then used as input into the second step,

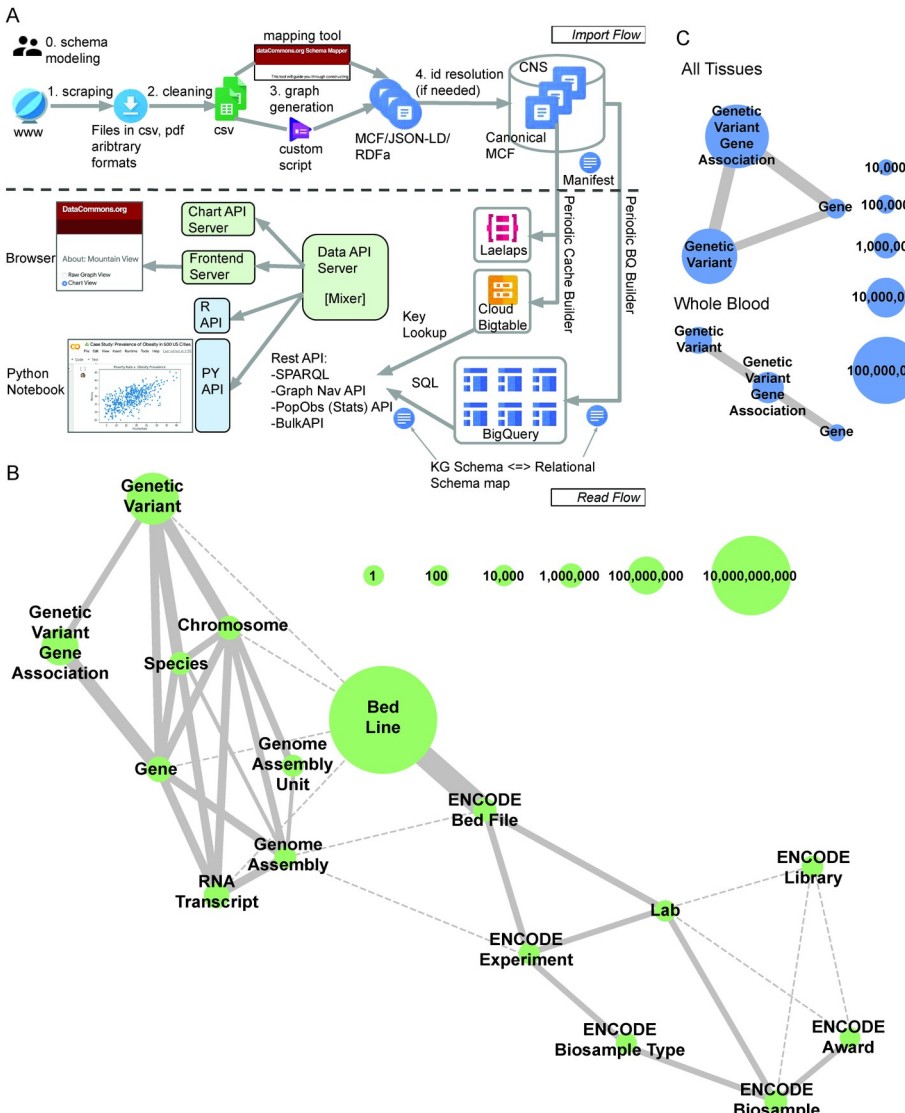

**Fig 1. Biomedical Data Commons is a knowledge graph that integrates multiple data types. A:** Workflow for cleaning, formatting, ingesting, and accessing data in the Google Biomedical Data Commons knowledge graph. **B:** Current state of the Google Biomedical Data Commons graph. The size of the node indicates the number of unique entities of that type in the graph. Solid edges depict explicit relationships between two entity types with edge width corresponding to the number of unique links between the entity types. Dashed line edges denote implicit relationships in the graph. **C:** A depiction of the subgraph of Biomedical Data Commons displaying the Gene, GeneticVariant, and GeneGeneticVariantAssociation nodes and edges in total (top panel) and the subset of which are reported as significantly associated in Whole Blood by GTEx (bottom panel). Node size and edge width correspond to the number of unique entities of that type and relationships between two entity types respectively. This represents how a user can use the Data Commons API to search and retrieve information contained in a subset of the graph in which they are interested.

which results in the output of gene regulatory network visualizations and a ranked list of genetic variants and genes that are prioritized by their level of connectivity in their local regulatory network. The output gene and genetic variant list generated by this pipeline can be used as input into BMDC to obtain additional information on these candidates and provide the user with a more comprehensive picture into their biomedical function.

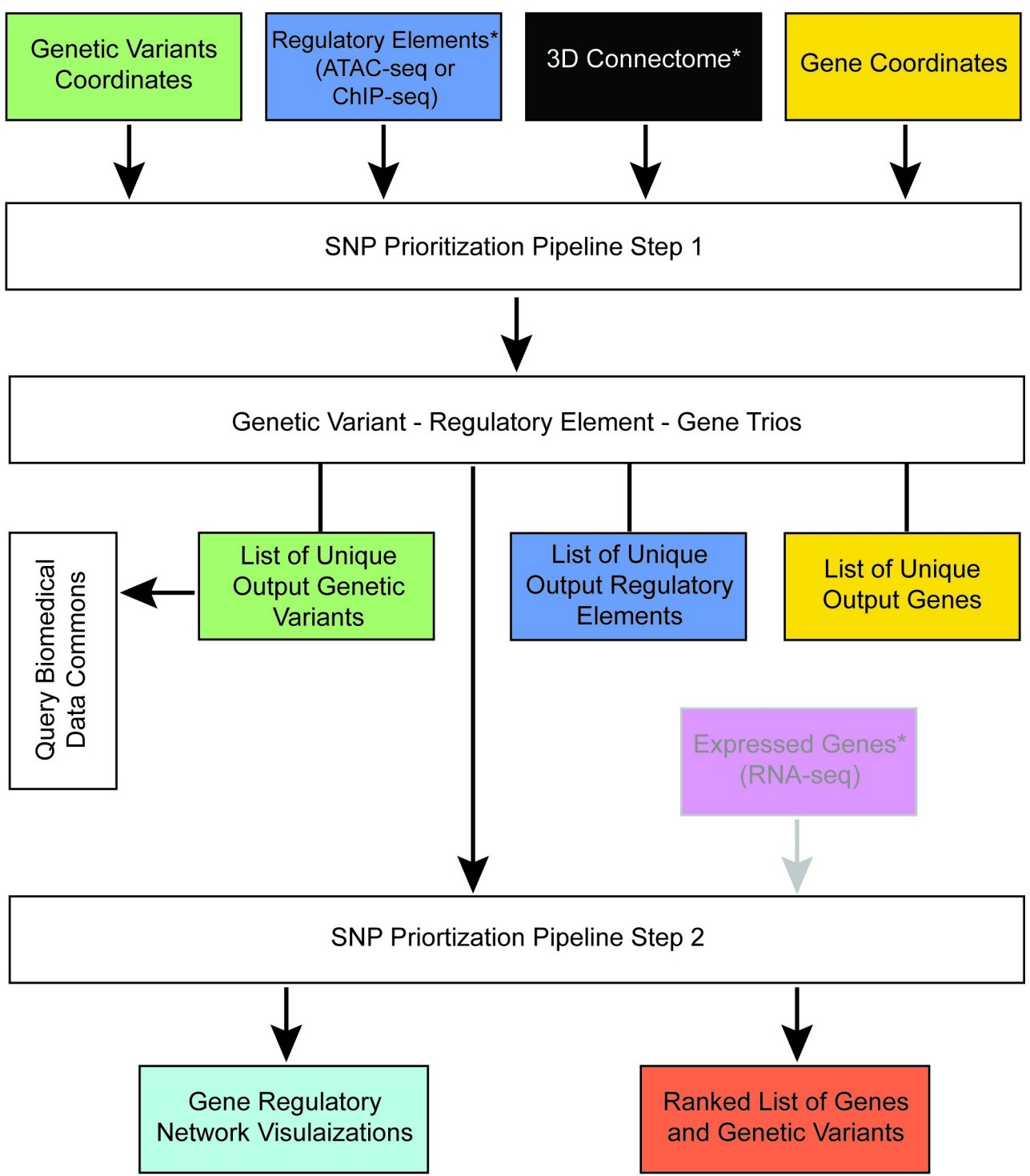

**Fig 2. Schematic of the SNP Prioritization Pipeline.** Genomic coordinates of genetic variants, regulatory elements, 3D connectome, genes are input into step one of the SNP Prioritization Pipeline. The output from Step 1 are associated genetic variants—regulatory elements—genes, which together formed trios. List of the unique genetic variants, regulatory elements, and genes that participate in these trios are also outputted. The gene list from step 1 of the SNP Prioritization Pipeline can be used as input for Biomedical Data Commons queries. The trios generated in step 1 are used as input into step 2 of the SNP Prioritization Pipeline along with optional input cell type-specific gene expression data. The output is visualizations of gene regulatory networks and a ranked list of genetic variants and genes. Input and output data at each step of the pipeline is color coded by type of data: genetic variants (green), regulatory elements (blue), genes (gold), 3D connectome (black), gene expression data (magenta), gene regulatory networks (turquoise), and ranked list of genetic variants and genes (orange). Optional input data is denoted by grey text, grey outline of the input box, and grey input arrow. *denotes input data that is cell type-specific and needs to belong to the same cell type of interest throughout the pipeline.

## SNP Prioritization Pipeline identifies non-coding and tissue-specific variants

A list of 267 previously published significant T1D GWAS variants and an additional 12,707 variants in linkage disequilibrium with these significant variants was compiled (S3 Fig).[29] Genetic variants in regulatory elements—open chromatin regions or transcription factor binding sites—in GM12878 B-lymphocyte cells were associated with gene targets using three-dimensional chromatin conformation data defined by H3K27ac HiChIP (Fig 3A). This identified 602 unique genetic variants participating in 8,682 unique genetic variant—regulatory element—gene trios with 473 unique gene targets (S3 Fig). The vast majority of these output candidate variants are of unknown clinical significance as recorded by ClinVar (Fig 3B). The output candidate variants are also enriched for non-coding regions compared to the input genetic variants as defined by the genomic location recorded in dbGAP (Fig 3C). This is as expected because the pipeline filters for variants in defined regulatory elements. ~18% of output candidate variants have previously been significantly associated by GTEx with at least one of their assigned gene targets in pancreas, thyroid, or whole blood (S2B Fig). Information on the clinical significance, genome location, and known significant gene associations of genetic variants was identified using the BMDC python and SPARQL API (Figs 3B and 3C, and S2A and S2B).

The pipeline was also run using cell type-specific regulatory elements and three-dimensional connectome data from 1° Naive T, Th17, and Treg cells. Comparison of the output genetic variants from each of these cell types reveals that the candidate variants identified were cell-type specific (Fig 3E, left panel). However, the gene targets identified by the pipeline were largely the same between the GM12878, and 1° Naive T, Th17, and Treg cells (Fig 3E, right panel).

Kegg pathways and gene ontology terms were used to provide further insight into the biological role of the pipeline identified gene targets of the output candidate variants using GM12878 cell type-specific input data. One of the top KEGG pathways is T1D and other top KEGG pathways are other autoimmune diseases (Fig 3F). The top gene ontology biological processes involve regulation of cytokine signaling and T cells (S4C Fig).

## Output candidate variants are common and not evolutionarily conserved

Using the BMDC python API, we observed the majority of output candidate variants identified by the pipeline (S3 Fig) to be common with a minor allele frequency (MAF) greater than 2% (Fig 3G). The providence for this data extracted from BMDC is dbGAP. The distribution of the minor allele frequency of output candidate variants was distinct for each ethnicity (S2D–S2H Fig). Of note, East Asians had a larger proportion of output candidate variants with a rare minor allele frequency of less than 2% compared to other ethnicities (S2G Fig). The vast majority of output candidate variants have low CADD scores, which are calculated largely using evolutionary conservation data alongside additional metrics (Fig 3H). This is suggestive that the variants are not evolutionarily conserved. The top transcription factor binding motifs in pipeline-identified regulatory elements are CTCF/Boris and NFkB (Fig 3I). Compared to the original input list of T1D-associated variants based on GWAS and linkage disequilibrium, there is an enrichment of output candidate variants in chr6 (Fig 3J). Chr6 contains the HLA locus that accounts for ~50% of the genetic contribution involved in early-onset T1D.[30]

## Using cohesin HiChIP data as input identifies a subset of the trios found using H3K27ac HiChIP input data

92.1% of trios identified using cohesin HiChIP three-dimensional chromatin conformation data were also identified using H3K27ac HiChIP data in GM12878 cells (S4A Fig). However,

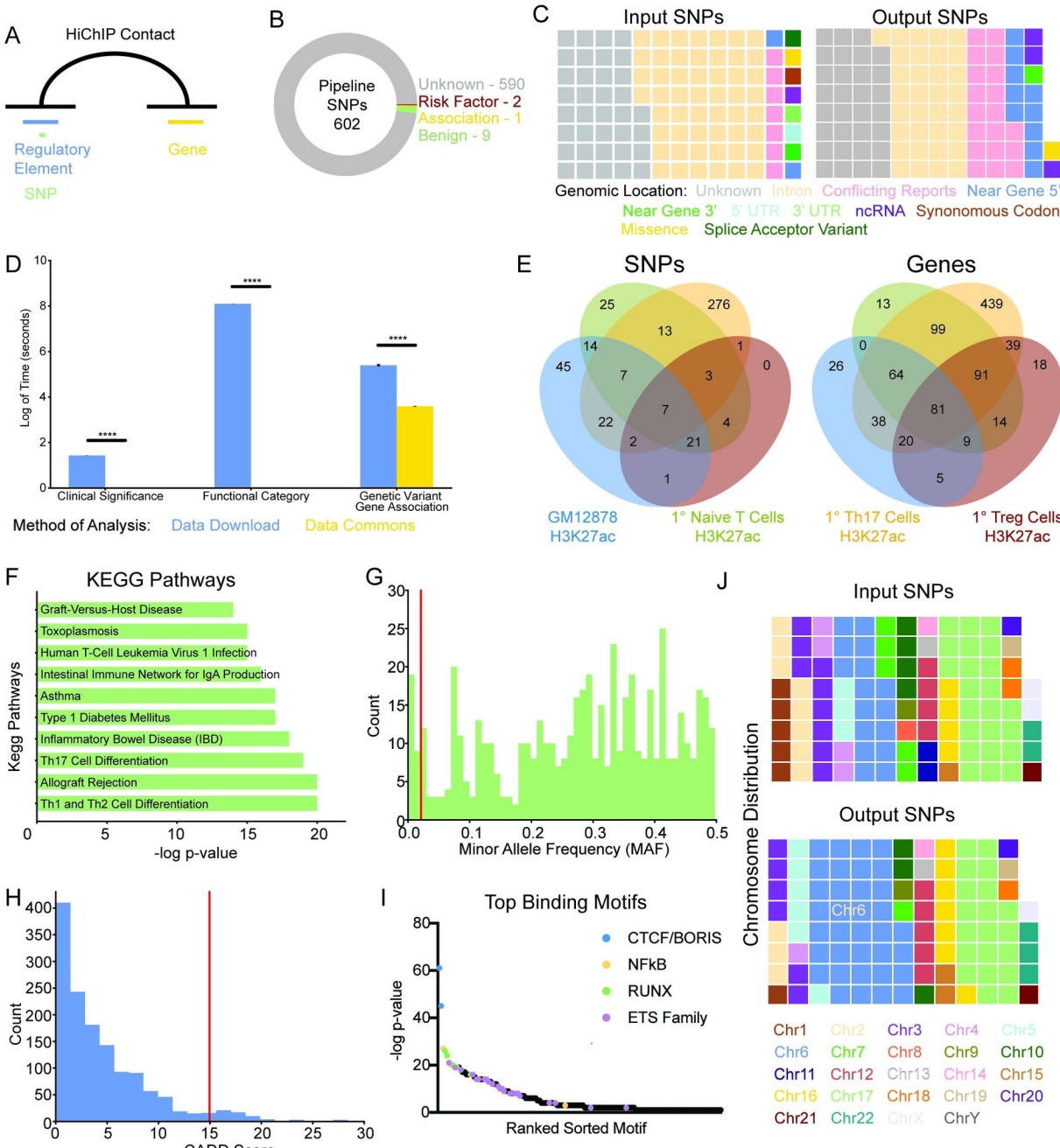

**Fig 3. Pipeline prioritizes common non-coding genetic variants that are cell type specific. A:** Overview of the pipeline that identifies genetic variants in regulatory elements of interest distally connected to genes via the three-dimensional chromatin conformation. **B:** Clinical significance of pipeline output genetic variants. **C:** Functional category of input and output genetic variants. **D:** The amount of time to run clinical significance, functional category, and significant gene association analyses using local data scientist approach involving data download (blue) or Data Commons (gold). **** p < 0.0001. **E:** Venn diagram of the overlap in genetic variants (left panel) and genes (right panel) identified by the pipeline using H3K27ac HiChIP and ATAC-seq datasets from GM12878 cells (green) and primary Naïve T (green), Th17 (gold), and Treg (burgundy) cells as input. **F:** KEGG pathways associated with the target genes of pipeline genetic variants. **G:** Histogram of the minor allele frequency of the pipeline genetic variants. Red line at 0.02 indicates common cutoff for uncommon genetic variants. **H:** Histogram of pipeline genetic variants CADD score with red line at a score of 15 indicating a common cutoff for deleterious variants. **I:** Scatter plot of the top binding motifs of pipeline identified regulatory elements. **J:** The chromosomal location of input and output genetic variants. Input SNPs refers to the original candidate list of 12,974 genetic variants that were reported significantly associated with T1D in the GWAS catalog or are in linkage disequilibrium. The Output SNPs refers to the 602 genetic variants that were identified as candidates by the pipeline.

cohesin HiChIP data identifies only a fraction (~8.4%) of the trios found using H3K27ac data. Gene Ontology Biological Processes and KEGG Pathways associated with output candidate variants identified using GM12878 cohesin HiChIP data are immune system-specific (S4B and S4C Fig). The top binding motifs in pipeline-identified regulatory elements with GM12878 cohesin HiChIP data are CTCF/Boris and NFkB—the same as in the pipeline-identified regulatory elements with GM12878 H3K27ac HiChIP data as input (S4D Fig). The proportion of output candidate variants that are a T1D GWAS significant variant versus a variant in linkage disequilibrium is slightly higher using cohesin HiChIP data (7.9%) compared to H3K27ac HiChIP data (3.5%; S4E Fig).

## Comparison to existing genetic risk score method

The T1D GRS2 genetic risk score method uses 67 genetic variants and was developed on patient data.[15] It has performed well particularly in the context of early-onset T1D. Our initial input list of T1D GWAS significant variants and those in linkage disequilibrium included 34/67 of the Sharp et al. variants. Of these 7 (~20.1%) were identified as significant using our prioritization pipeline with GM12878 input data. An additional 3 unique genetic variants were identified using 1° Naive or Th17 input data. Of note, rs3024505, whose gene target is IL19 and other nearby genes, was identified using input data from all three of these cell types. Numerous previous GWAS studies of autoimmune diseases have identified rs3024505 as significantly associated with multiple autoimmune diseases including T1D, inflammatory bowel disease, Behçet's disease, and others.[31–33] Furthermore, a number of the genetic variants identified in this study were nearby Sharp et al. variants and regulated the same gene targets including in the IL27 and UBASH3A loci (S5 Fig).[15] In addition, 24/40 Sharp et al. gene targets were identified using our pipeline with GM12878 input data.

## There are four clusters of gene regulatory network structures

The pipeline produced genetic variants—gene regulatory networks. This was done by representing genetic variants and their gene targets identified by the first part of the pipeline as a bipartite graph. This results in a graph composed of multiple components, each representing the gene regulatory networks at distinct loci. These graph components were converted to a matrix representation to enable machine learning on the gene regulatory networks. To identify if there were common network structures, K-means clustering was performed on the principal component analysis (PCA) of the gene regulatory networks. This identified four clusters representing distinct gene regulatory network structures (S6B and S6C Fig). The one that separated the most was a dense cluster populated only by the Major Histocompatibility Class II (MHC-II) locus (Dense; n = 1). Another cluster contained components in which each node only had a few connections to other nodes in the graph and did not contain an obvious focal point (Chain; n = 2). A third cluster had multiple obvious focal points (Multi-Focus; n = 9). The final cluster contained gene regulatory networks with simple structures that tended to have a single focal point (Simple; n = 27).

## Centrality analysis identifies HLA-DQB1 and HLA-DRB1 as top candidates

Centrality, a measure of the important nodes of the graph, was used to identify key nodes in the gene regulatory networks to recommend for further biological investigation. Closeness centrality, a calculation of the sum length of the shortest path between that node and all nodes in the graph, created the best distribution of centrality scores compared to alternative centrality measures degree and betweenness centrality (S6A Fig). The top candidate gene regulatory

network was the 2.2 Mbp MHC-II locus (Fig 4A). This was selected because it contained the first ranked genetic variant by closeness centrality score of the pipeline output list of ranked genetic variants and genes (S3 Table). Furthermore, it was the most dense gene regulatory network and contained the most connected genes (HLA-DQB1 and HLA-DRB1) as well based on centrality score.

The top candidates within the HLA locus were HLA-DQB1 and HLA-DRB1, which together form the haplotype DR4—one of two haplotypes with the strongest association with T1D (Figs 4B and S7A).[34] These genes are strongly connected by chromatin folding associated with all types of investigated regulatory elements containing one or more T1D-associated genetic variants (Figs 4C and S7B). The top genetic variant is novel rs14004 and it serves as a major connection point between the highly connected 1 Mbp HLA locus and the region 1.2 Mbp upstream (Figs 4B and S7A).

## Chromatin connectivity in HLA locus is B-lymphocyte-specific

The chromatin connectivity in the HLA locus between pipeline-identified candidate genetic variants and their gene targets is cell-type specific (Fig 4D and 4E). Only a limited number of these connections were observed in 1° Treg cells. The HLA-DQB1 transcription start site (TSS), HLA-DRB1 TSS, and rs14004 are all strongly connected with each other as well as rs9986640 (Fig 5A and 5B). rs14004 and rs9986640 are both located in BCL11A, RAD21, and STAT5 ChIP-seq binding sites (Fig 5A). rs14004 is additionally in a TCF3 binding site, and rs9986640 is additionally in a CTCF, PAX5, SMC3, and SPI1 binding sites. HLA-DQB1 and HLA-DRB1 are associated with 98 genetic variants by the pipeline (S4 Table). rs14004 is a common allele whereas rs9986640 is a rare alle (Fig 5C).

## IL2RA is identified as the top candidate for within its locus

To provide insight into how the pipeline performed at another well studied locus in the context of Type 1 Diabetes, we focused on the IL2RA locus. The pipeline identified the IL2RA as the top candidate within the locus by closeness centrality of its gene regulatory network (Fig 6A and 6B). IL2RA is strongly connected via chromatin folding to multiple types of regulatory elements containing T1D-associated genetic variants (Fig 6C). The connectivity within the IL2RA locus of between pipeline-identified genetic variants and gene targets is distinct between GM12878 and 1° Treg Cells (Fig 6D and 6E). The top genetic variant candidate in the IL2RA locus is rs61839660, which is located in a BCL11A, EBF1, IKZF1, MYB, PAX5, SPI1, and STAT5 ChIP-seq binding site (Figs 6B and 6C, and 7A). rs61839660 and IL2RA TSS are both connected with the novel variant rs198390 (Fig 7A and 7B). Both variants and the IL2RA TSS are also connected to the IL15RA TSS. rs61839660 is a rare allele, whereas rs198390 is a common allele (Fig 7C).

## Ikaros family identified as candidates

Finally, we performed a deeper dive into IKZF3 and IKZF1 to provide additional example gene regulatory networks produced by the SNP Prioritization Pipeline. These loci were chosen because they belong to the same transcription factor family and are known to be important in B-lymphocyte development with IKZF1 a critical transcription factor.[35] They are also implicated in early-onset Type 1 Diabetes, but their mechanism of action remains unclear.[36,37] Here, we use the gene regulatory networks produced by the SNP Prioritization Pipeline to generate hypotheses regarding the mechanism by which genetic variance may disrupt gene expression in these loci and contribute to Type 1 Diabetes (S8A–S8C and S9A–S9C Figs). IKZF3 is in a much more dense and complex gene regulatory region than IKZF1 (S8B, S8C, S9B and S9C

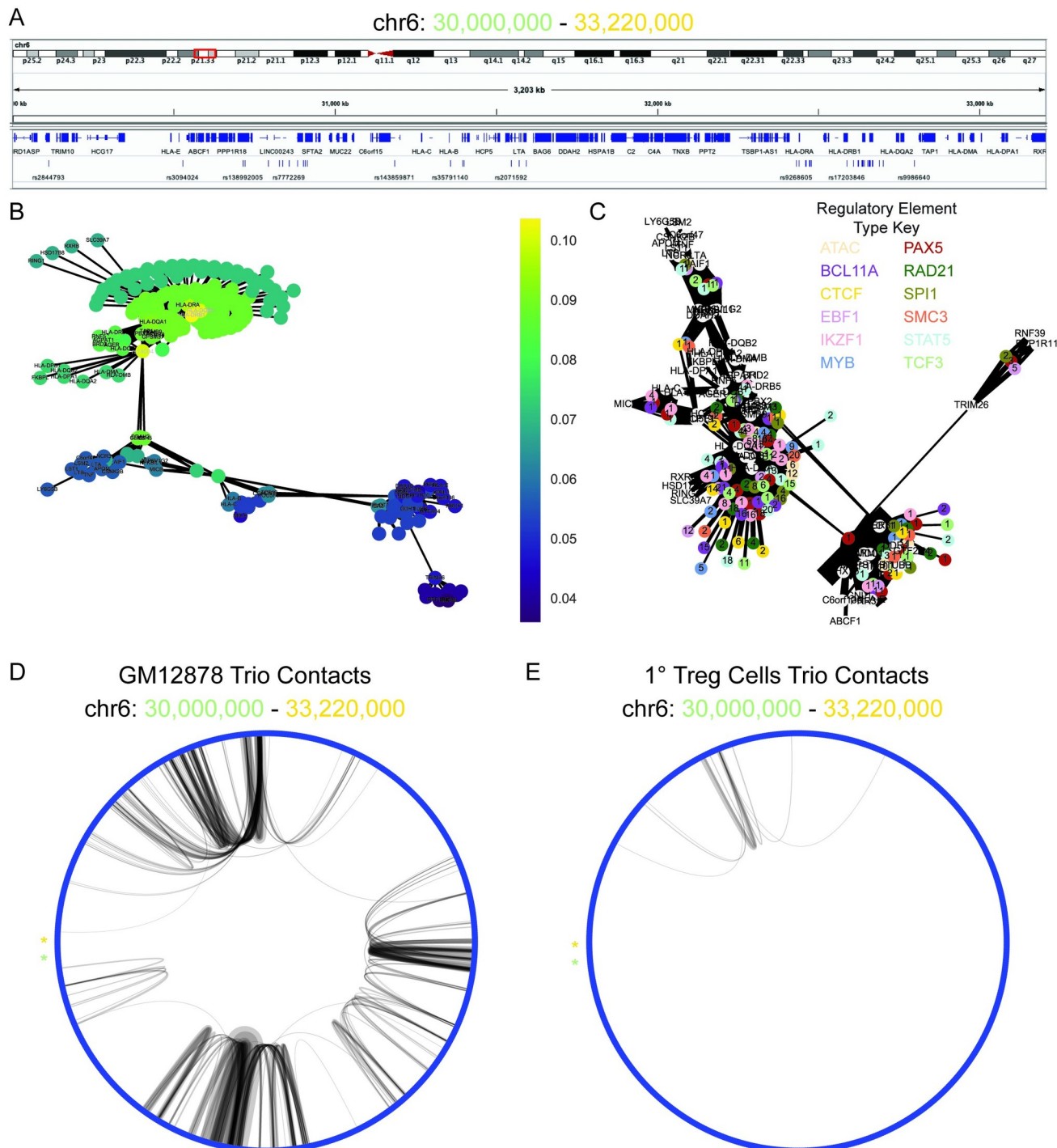

**Fig 4. HLA-DRB1 and HLA-DRQ1 are top pipeline identified candidates for T1D-associated genes in the HLA locus. A:** Visualization of the HLA component of interconnected pipeline genetic variant–regulatory element–gene trios (chr6: 30,000,000–33,220,000). **B:** Bipartite graph of the HLA component with gene and genetic variants as nodes and chromatin connections as edges. Node color indicates closeness centrality score with gold being most connected and purple being least connected nodes in the graph. Gene nodes are labeled, and genetic variant nodes are unlabeled. **C:** Bipartite graph of HLA component with gene and regulatory elements as nodes and chromatin connects as edges. Gene nodes are labeled and white. Regulatory element nodes are colored by type and labeled by the number of unique genetic variants contained in the regulatory element. The width of edges indicates connectivity strength as indicated by the number of unique HiChIP reads. **D:** Circos plot of the chromatin connectivity at 5 kb resolution in the HLA locus. The nodes are sections of the genome and the edges are the chromatin connectivity with the width indicating connectivity strength. An asterisk labels the starting (chr6: 30,000,000; green) and terminating (chr6: 33,220,000; gold) nodes of the plot. GM12878 (left panel) and Treg (right panel) pipeline trio contacts are visualized.

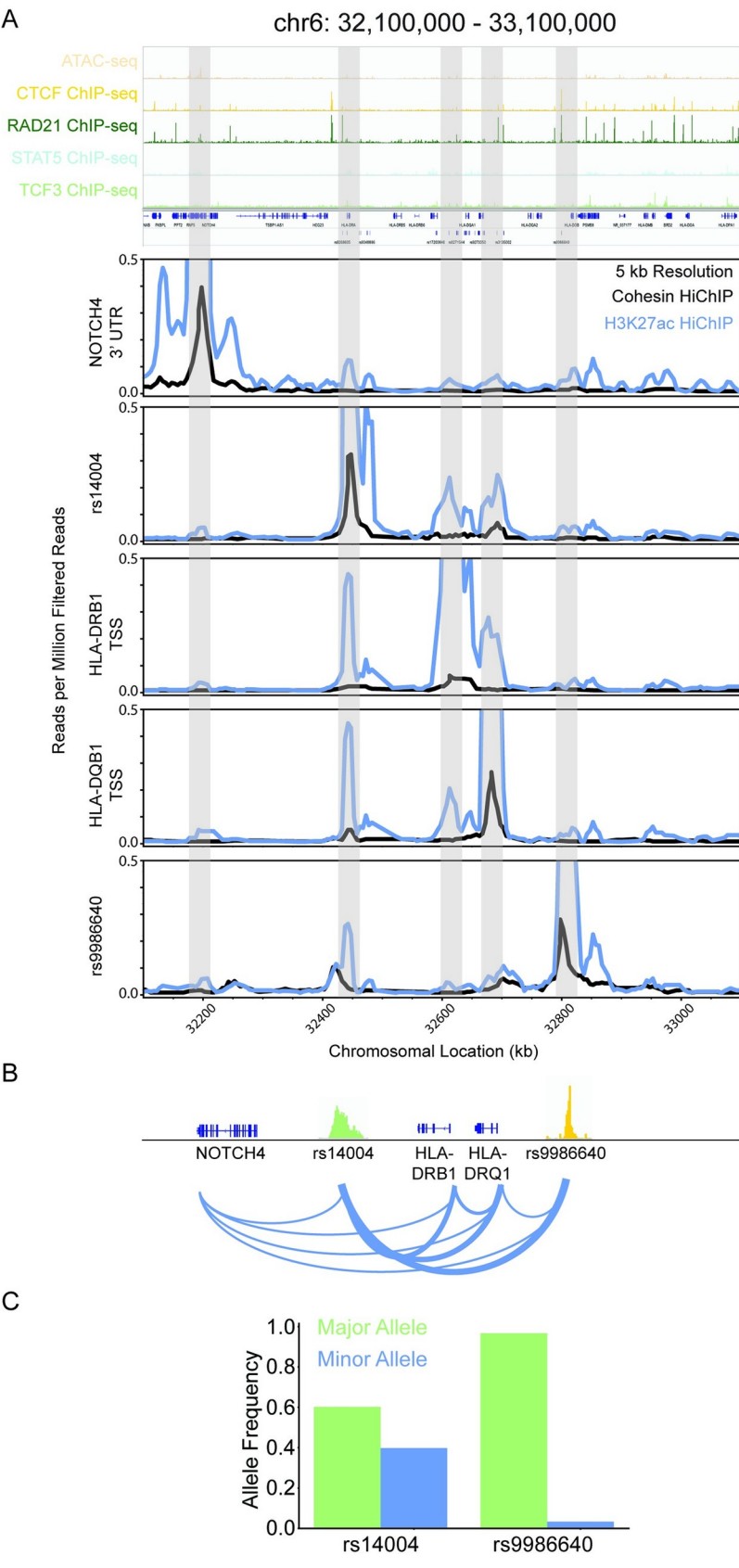

**Fig 5. Novel genetic variant rs14004 is a candidate for gene expression regulation of HLA-DRB1 and HLA-DQB1.**
**A:** Visualization of the portion of the genome that interacts with HLA-DRB1 and HLA-DQB1 (chr6: 32,100,000–
33,100,000). ATAC-seq and CTCF, RAD21, STAT5, and TCF3 ChIP-seq raw read visualization (top panel). Cohesin
(black) and H3K27ac (blue) HiChIP raw reads virtual 4C plots centered on the Notch 3' UTR, rs14004, HLA-DRB1
TSS, HLA-DQB1 TSS, and rs9986640 (bottom panel). **B:** Schematic of the chromatin connectivity between the genetic
variants and the genes as represented by the raw data for chr6: 32,100,000–33,100,000. **C:** Major (green) and minor
(blue) allele frequencies from 1000 Genome Project for rs14004 and rs9986640. **D:** Primary whole blood RPKM values
for NOTCH4, HLA-DRB1, and HLA-DQB1 of healthy and Type 1 Diabetes patients.

Figs). The chromatin connectivity between pipeline-identified genetic variants and their gene
targets in both of these loci are B-lymphocyte specific with no such connectivity observed in 1˚
Treg cells, which is to be expected because IKZF1/3 are known to specifically function in B-
lymphocytes (S8D, S8E, S9D and S9E Figs).

## Discussion

The lack of a publicly available, queryable central biomedical database hobbles biomedical
research by increasing the time, resources, and level of expertise needed to evaluate publicly
available biomedical data. We address this by developing BMDC, a knowledge graph that inte-
grates data from multiple sources into a single queryable format. To demonstrate its useful-
ness, we applied it to aid our understanding of the role of B-lymphocytes in T1D. Currently,
there are limited methods for identifying the role of non-coding genetic variants in specific
cell types and disambiguating polymorphism. To address this, we created a pipeline that lever-
ages the cell type-specific enhancer and three-dimensional chromatin structure in a cell to pri-
oritize genetic variants for biological validation. It prioritized non-coding genetic variants that
are cell type-specific enhancers providing insight into the role of genetic variance and B-lym-
phocytes in T1D. We used BMDC to extract publicly available information on pipeline-identi-
fied variants to increase interpretation of their function. Together, this leverages publicly
available big data and data integration to provide insight into the role of non-coding variants
and polymorphism in complex disease.

BMDC removes inefficiencies in data cleaning and democratizes biomedical data by serving
as a central database accessible from datacommons.org. Data Commons is open—any user can
contribute datasets or build applications powered by the graph using our API. The biomedical
data is organized in an open-source knowledge graph in accordance with the Findability,
Accessibility, Interoperability, and Reusability (FAIR) data principles.[38] As BMDC is com-
posed of cleaned data integrated from multiple data sources into a single format this dramati-
cally reduces the time that users spend on cleaning data. This democratizes the data by
increasing the ease of data sharing and reducing the programming ability needed to evaluate
the data thereby decreasing the barrier to entry for conducting biomedical analyses.

BMDC is the first large scale, publicly available, and community-based platform knowledge
graph in the biomedical space. Another major effort in this space is The Biomedical Data
Translator Consortium.[39] This consortium is building a platform that supports the creation
of knowledge graphs that are focused on a defined area of biomedicine like a given disease. It
is a federated system that requires application to have access to specific graphs built by the
group. It remains primarily in use by consortium members, and it's unclear when the platform
will be made broadly available to the biomedical community. Another biomedical knowledge
graph that has been built is the Scalable Precision Medicine Oriented Knowledge Engine
(SPOKE), an effort out of University of California, San Francisco.[40] The biggest difference
between BMDC and SPOKE is accessibility and size. SPOKE remains a private graph and

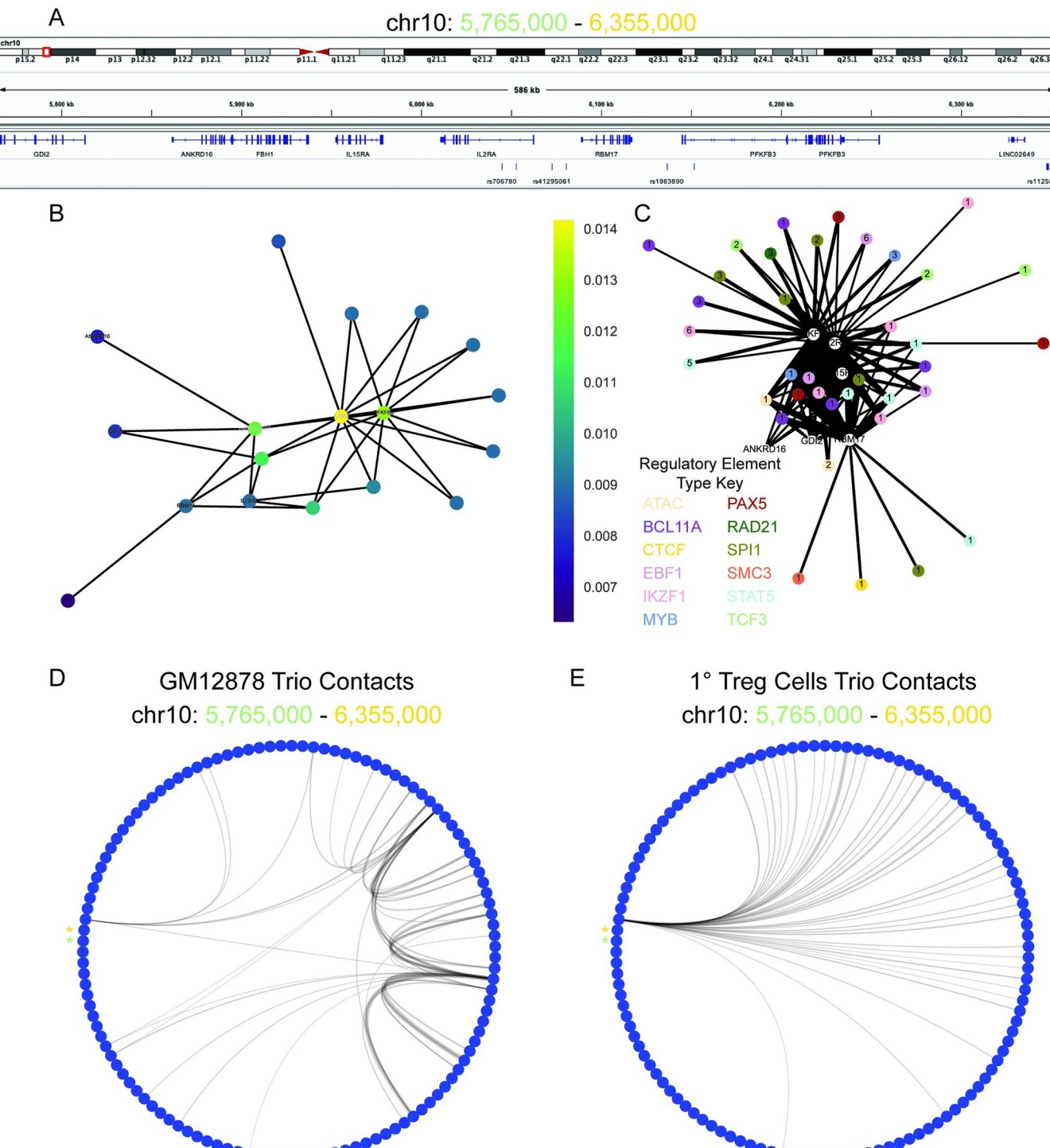

**Fig 6. IL2RA is a top pipeline identified candidate. A:** Visualization of the IL2RA component of interconnected pipeline genetic variant–regulatory element–gene trios (chr10: 5,765,000–6,355,000). **B:** Bipartite graph of the IL2RA component with gene and genetic variants as nodes and chromatin connections as edges. Node color indicates closeness centrality score with gold being most connected and purple being least connected nodes in the graph. Gene nodes are labeled, and genetic variant nodes are unlabeled. **C:** Bipartite graph of IL2RA component with gene and regulatory elements as nodes and chromatin connects as edges. Gene nodes are labeled and white. Regulatory element nodes are colored by type and labeled by the number of unique genetic variants contained in the regulatory element. The width of edges indicates connectivity strength as indicated by the number of unique HiChIP reads. **D:** Circos plot of the chromatin connectivity at 5 kb resolution in the IL2RA locus. The nodes are sections of the genome and the edges are the chromatin connectivity with the width indicating connectivity strength. An asterisk labels the starting (chr10: 5,765,000; green) and terminating (chr10: 6,355,000; gold) nodes of the plot. GM12878 (left panel) and Treg (right panel) pipeline trio contacts are visualized.

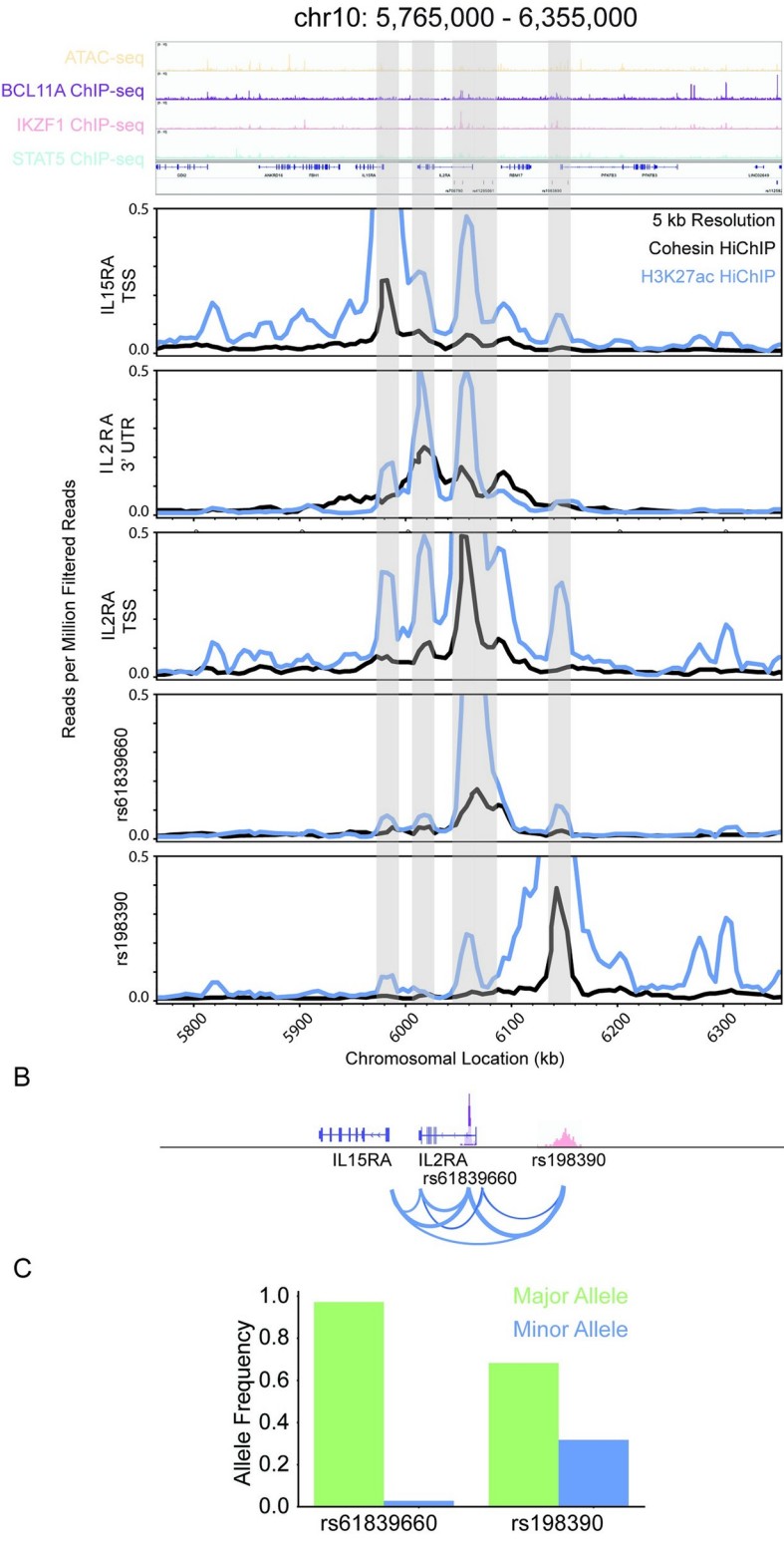

**Fig 7. rs61839660 identified as a candidate for gene expression regulation of IL2RA and IL15RA. A:** Visualization of the portion of the IL2RA gene regulatory network (chr10: 5,765,000–6,355,000). ATAC-seq and BCL11A, IKZF1, and STAT5 ChIP-seq raw read visualization (top panel). Cohesin (black) and H3K27ac (blue) HiChIP raw reads virtual 4C plots centered on the IL15RA TSS, IL2RA 3' UTR, IL2RA TSS, rs61839660, and rs198390 (bottom panel). **B:** Schematic of the chromatin connectivity between the genetic variants and the genes as represented by the raw data for

chr10: 5,765,000–6,355,000. **C:** Major (green) and minor (blue) allele frequencies from 1000 Genome Project for rs61839660, and rs1983900.

contains only 47,000 nodes and 2.25 million edges making it four orders of magnitude smaller than the current BMDC graph.

Here, we demonstrate a use case of how BMDC can be applied to aid in the prioritization of genetic variants associated with complex phenotypes. Our method prioritizes non-coding variants in a cell type-specific manner by using the epigenomic state of the cell. It provides insight into potential mechanisms of action via the regulatory element with which the genetic variant is located. It also builds gene regulatory networks which provide insight into potential polymorphic and/or polygenetic effects. Furthermore, it creates a locus-specific score to prioritize alleles in complex loci.

We applied our pipeline to prioritize genetic variants involved with regulation of B-lymphocyte function in T1D, one of most-studied childhood diseases. B-lymphocytes appear to play a role in early-onset diabetes and their involvement in the pathogenesis is associated with more aggressive disease progression.[41,42] In addition, immortalized B-lymphocyte cell line GM12878 has been extensively characterized and has a wealth of publicly available epigenetic datasets available making it an excellent model for testing our method. Our method identified a substantial fraction of the variants identified by Sharp et al. GRS2 method and the majority of the same gene targets.[15] In the case of rs5763779, Sharp et al. assigned it to the nearest gene HORMAD2, whereas our method associated it with oncostatin M. Albiero et al. previously showed that excess levels of oncostatin M inhibit mobilization of stem cells into the peripheral blood in T1D patients thus offering a potential mechanism by which rs5763779 contributes to the manifestation of T1D.[43] This showcases the value that our method offers by providing a way to assign non-coding regions to distal gene targets by chromatin looping.

The densest loci identified by our pipeline was the MHC-II locus which accounts for ~50% of the genetic contribution involved in early-onset T1D.[30] The most connected entities in the locus were HLA-DQB1 and HLA-DRB1, which together can form the haplotype DR4, one of two haplotypes which conveys the highest risk for T1D.[34] In addition, Inshaw et al. recently reported HLA-DQB1 and HLA-DRB1 haplotypes DR15-DQ6 (DRB1*15:01-DQB1*06:02) and DR7-DQ3 (DRB1*07:01-DQB1*03:03) were least common in children that were under the age of 7 at the time of diagnosis with T1D.[36]

The top genetic variant identified by our analyses rs14004, which is in a RAD21 binding site, interacts with the rare allele rs9986640, which is in a strong CTCF binding site that colocalizes with cohesin. Interaction of this CTCF site has been shown by 3C to connect HLA-DQB1 and HLA-DRB1 promoters in an interferon-gamma-dependent manner in Burkett's lymphoma B-cell line Raji.[44] Although, no individual knockdown or knockout of this cite has been made, siRNA knockdown of CTCF leads to reduced expression of the entire MHC-II locus and interferon-gamma leads to increased CTCF-dependent looping and is associated with increased HLA-DQB1 and HLA-DRB1 expression. The regulatory elements that contain rs14004 or rs9986640 both interact with the two superenhancers in the intergenic region of HLA-DQB1 and HLA-DRB1. Loss of these superenhancers leads to decreased connectivity in the MHC-II locus and lower expression levels of HLA-DQB1 and HLA-DRB1.[45] Together this suggests an important role of chromatin connectivity in the HLA-locus for regulation of gene expression and suggests a mechanism of action by which genetic variance disrupts looping of regulatory elements containing rs14004 or rs998860 leading to misregulation of HLA-DQB1 and HLA-DRB1.

IL2RA was another locus that was identified by our pipeline with rs61839660 in the CaRE4 IL2RA enhancer identified as the locus's top genetic variant candidate. Previously, Simenov

et al. demonstrated that the presence of the minor allele led to decreased enhancer activity and IL2RA expression in Jurkat CD4+ T cells.[46] Deletion of the CaRE4 enhancer in mice leads to protection against development of T1D even when the mice are treated with an immunosti-mulating anti-PD1 checkpoint inhibitor.[47] This provides validation in mice and humans with how the pipeline-identified rs61839660 trio works. rs61839660 affects the function of the CaRE4 enhancer in which it resides, which in turn leads to decreased IL2RA expression and subsequently protects against T1D. This provides confidence in using this method to prioritize genetic variants for biological validation in complex diseases.

This study contains limitations that are worth addressing in future research. Our analyses in this study are limited to B and T-lymphocytes. It is possible that genetic variants identified in this study are also involved in gene regulation in other cell types involved in Type 1 Diabetes manifestation and disease progression including dendrites and B-islet cells. We recommend that the pipeline be run with cell type-specific input data for all cell types involved in a given disease to provide a more complete picture as to which genetic variants may be involved in regulating which genes in which cells. This would provide the most insight into how genetic variants may be impacting the function of multiple cell types. In addition, most of this study was focused on the immortal B lymphocyte cell line GM12878. We chose this cell line because we wanted to validate our approach in a well-studied cell line that had all the needed multidi-mensional 'omics data readily available. A limitation of this is that there may be some aspects of the epigenome that are unique to B-lymphocytes that are not present in 1° B-lymphocytes. It would be interesting to compare the results from 1° B-lymphocyte data.

The multiple candidates generated by the pipeline have already been identified and biologi-cally validated in the literature in both mice and humans. In particular, the CaRE4 enhancer mechanism of action for regulation of IL2RA expression is exactly what is expected based on the insight provided by the IL2RA regulatory network produced by the pipeline.[45,46] This provides confidence that our pipeline is identifying candidate genetic variants and regulatory elements that are worth investigating their function *in vivo*. The pipeline also identified many loci that are associated with Type 1 Diabetes and B-lymphocyte development but are much less studied. We suggest that follow-up studies should be conducted into the role of the highly connected pipeline-identified regulatory regions in regulating their gene targets.

Here, we demonstrate a single use case of BMDC to understand the role of non-coding genetic variants in complex diseases. However, BMDC can be used to address a wide range of biomedical questions dependent on big data analyses. The open-source nature of BMDC allows for organic community growth and flexibility in use and scope of BMDC to advance biomedical research.

## Conclusions

We have built BMDC, a platform that facilitates easy wide-spread sharing and analysis of biomed-ical data. We demonstrate how BMDC can be leveraged in the use case of prioritization of cell type specific disease alleles associated with complex phenotypes. The SNP prioritization pipeline provides insight into the potential polymorphic effects and mechanism of action by which non-coding genetic variants regulate gene expression. The function of top genetic variant candidates identified by this method have been validated in the literature in both mice and humans.

## Methods

### Processing of ChIP-seq and ATAC-seq data

Sequence alignment to hg38 was performed using bowtie.[48] for ATAC-seq (parameters: -p 24 -S -m 1 -X 2000). The following was completed for both ATAC-seq and ChIP-seq datasets.

Samtools was used to remove PCR duplicates and mitochondrial DNA (for ATAC-seq data-sets) from aligned reads.[49] Peak calling was carried out with MACS2.[50] using default set-tings with a p-value cutoff of 0.05. To filter out non-reproducible peaks, called peaks from biological replicates were processed through the Irreproducible Discovery Rate (IDR) frame-work implemented in R with a p-value cutoff of 0.01.[51]

## Processing of HiChIP Data

HiChIP paired end reads were aligned to hg38 using HiC-Pro.[52] Duplicate reads were removed, assigned to MboI restriction fragments, filtered for valid interactions, and then used to generate binned interaction matrices of both 5 kb resolution. High confidence contacts (defined as counts $\geq 10$, FDR $< 0.01$) using the contact caller FitHiChIP with default settings. [53] These high confidence contacts were used in the subsequent analyses including visualizing contacts by using interaction matrices created by HiC-Pro to create Virtual 4C profiles through a custom python script deploying the matplotlib library.

## Biomedical Data Commons schema development

Biomedical Data Commons is built on top of schema.org. The data model for both schema.org and Data Commons covers entities and the relationships between entities. This is organized as a set of entities that are arranged in a multiple inheritance hierarchy allowing each type to potentially be the subclass of multiple entities. Each entity has a set of properties. These prop-erties can have one or more domains meaning that they can be instances for any of these enti-ties. They can also have one or more ranges, and the values of each property should be one of those specified types. Any property that has a limited set of values is converted to schema.org class Enumeration. In addition, any observations or statistical variables of a population are represented as schema.org classes StatisticalPopulation and Observation and/or Data Com-mons class StatisticalVariable. An example of which would be the observed allele frequency of a given genetic variant in a given study such as the 1000 Genomes Project. In this case the Sta-tisticalPopulation would be defined as the specific allele for which the frequency observation is being observed and the Observation would contain the frequency recorded in that study along with additional information on the observation including the source and date observed.

Similarly, to schema.org, Data Commons is a collaborative platform that is open for com-munity development. Schema is extended as needed to represent a dataset, preserving the orig-inal encodings from the data source as much as possible, while maintaining consistency with the existing Data Commons schema. In these cases, any acronyms in property names that are domain specific are expanded to increase user accessibility. The current schema unique to Bio-medical Data Commons is available in github repository https://github.com/datacommonsorg/schema/tree/main/biomedical_schema or viewed on the datacommons.org browser. The Biomedical Data Commons specific schema as of 4/9/21 can also be viewed in S6–S8 Tables. More on the Data Commons data model can be found in our documentation https://docs.datacommons.org/data_model.html.

## Ingesting data into Biomedical Data Commons

Documentation on Data Commons can be viewed at https://docs.datacommons.org/. Data was downloaded directly from the original database and parsed into single property–value pairs that are associated with a single entity. The data was then represented in MCFs (http://www.guha.com/mcf/mcf_spec.html), which were then ingested into the Biomedical Data Commons knowledge graph (Figs 1A and S1). The providence for each property—value pairs is maintained and can be queried upon.

To enable joins across datasets, shared entities need to be reconciled. This was achieved through careful generation of dcid, a global Data Commons ID for each entity. The dcid was systematically crafted for each entity type by creating it from the most comprehensive and specific identifier or classification system for that type of entity. In cases that this identifier was not provided in the imported data, then the data was mapped to the appropriate identifier upon MCF conversion. This enables resolution of data from multiple databases into a single entity in the knowledge graph. Custom scripts for converting the original datasets to MCF can be found in github repositiory https://github.com/datacommonsorg/data/tree/master/scripts/biomedical.

## Biomedical Data Commons knowledge graph data storage and access

The generated MCF files are imported using the flow described in Fig 1A. They are stored in Google Cloud Bigtable and BigQuery. The data is accessible at datacommons.org and using the Google Data Commons API. Instructions for setting up the Google Data Commons API can be found at http://docs.datacommons.org/api/setup.html. Entities and triples of each type were quantified by performing SQL queries on the knowledge graph stored in BigQuery (S1 and S2 Tables).

## Generating input Type 1 Diabetes genetic variant list

Normalized significant genetic variants associated with Type 1 Diabetes were obtained from GWAS Catalog trait Type 1 Diabetes Mellitus: EFO_0001359.[29] The genetic variants in linkage disequilibrium with these significant genetic variants for Africans, Americans, East Asians, Europeans, and South Asians were obtained using SNiPA.[54] The linkage disequilibrium analysis was done in genome assembly GRCh37, variant set 1000 Genomes Phase 3 version 5, genome annotation Ensembl 87, and linkage disequilibrium threshold 0.7. The union of significant genetic variants and genetic variants in linkage disequilibrium with them in one or more populations to create the input genetic variant list. The genomic coordinates of these genetic variants were converted into genome assembly hg38 using the UCSC Genome Browser liftover tool.[55] This generated the Type 1 Diabetes-associated genetic variants list was used as input into the SNP Prioritization Pipeline in all analyses, regardless of cell type.

## Pipeline for prioritizing non-coding genetic variants

The pipeline requires the following input files: file path of the file containing chromatin contacts within a given cell, file path of the bed file of the regulatory element of interest, file path of the bed file of the genetic variants of interest, file path of the bed 5+ file of genes, and file path of the output file. There is an optional argument of the setting the wing-size—the number of base-pairs upstream and downstream considered extensions of the gene–with the default parameter set to 5,000 base pairs. Here we used H3K27ac HiChIP and Smc1a HiChIP datasets as input for the chromatin contacts and ATAC-seq and ChIP-seq datasets as input for the regulatory elements. Only one type of regulatory element dataset can be used as input into the algorithm for any given run.

The algorithm involves two steps. The first step is to define the associated regulatory elements—genetic variants—gene "trios". The second step uses these trios as input to build bipartite graphs of the gene regulatory networks defined by these trios. These are then used to perform centrality to produce a ranked list of genetic variant candidates for biological validation.

First, the algorithm filters regulatory elements for those that are participating in chromatin looping. Second, it filters for regulatory elements containing one or more genetic variants

thereby associating the regulatory elements with the genetic variants due to colocalization. Finally, it associates the paired regulatory element and genetic variant to distal gene targets by chromatin looping (i.e. it filters for pairs in which the other bin of the chromatin loop contains a gene). This forms an output trio composed of an associated regulatory element—genetic variant—gene. These are written to an output file in which each line is the genetic position and name of the unique regulatory element, genetic variant, and gene trio along with the chromatin contact. The chromatin contact recorded is the one that associates the regulatory element and genetic variant with the gene target due to the three-dimensional configuration of the DNA bringing them into close physical proximity with each other.

If the SNP Prioritization Pipeline algorithm is performed with multiple different types of regulatory elements as input, then the output files are combined to create one file containing all unique trios and their associated chromatin contacts. This is then used as input into the next step of the algorithm. First the genetic variants and their gene targets were filtered for the pairs for which the gene is known to be expressed in the cell type of interest. This filtering step is optional. Gene regulatory networks were then built by representing genetic variants and their gene targets identified in the first part of the pipeline as a bipartite graph. This graph is composed of multiple components, each of which represent the gene regulatory networks of individual loci. Each component is then analyzed using closeness centrality and the visualization is outputted. In addition, a ranked list based on closeness centrality score for genes and genetic variants.

The higher the closeness centrality score the more connected that node in the graph and the more likely that it plays a critical role in that regulatory network. Use of degree, betweenness, page rank and eigenvector centrality as the method by which to rank targets were also considered (S6A Fig). Closeness centrality was selected because it provided the most dynamic range in values for nodes in the gene regulatory networks.

## Pipeline deployment for cell type-specific analysis

The SNP Prioritization Pipeline was run using immortal B-lymphocyte cell line GM12878 H3K27ac HiChIP data along with ATAC-seq and ChIP-seq (BCL11A, CTCF, EBF1, IKZF1, MYB, PAX5, RAD21, SMC3, SPI1, STAT5, TCF3).[56–58] These are the data used for all the analyses in the paper unless otherwise specified. We also ran the algorithm substituting GM12878 cohesin HiChIP data as input for the chromatin data to compare the results of the analysis using H3K27ac versus cohesin HiChIP.[59] The pipeline was also run using primary cell Naïve T, Th17, and Treg Cells H3K27ac HiChIP and ATAC-seq data.[58] Along with the GM12878 H3K27ac HiChIP and ATAC-seq output, the resulting output genetic variant and target genes identified by the pipeline for each cell type were compared using a 4-way venn diagram.

## Motif discovery, KEGG Pathway, gene ontology, and CADD score

*De novo* motif discovery was performed using Homer findMotifsGenome.pl command with– size 200 as a parameter (version 4.8). Results were visualized in a scatter plot using GraphPad. KEGG Pathway and gene ontology biological process terms analysis was performed using Enrichr and results were visualized in bar charts.[60] CADD scores were obtained using https://cadd.gs.washington.edu/ and were visualized in a histogram.[9]

## Genetic variant clinical significance, functional category, minor allele frequency, and significant gene associations

For an input list of genetic variants, data on the clinical significance and functional category were obtained by querying Google Data Commons using the python API and the original

providence of the data is ClinVar and dbSNP respectively.[61,62] Data was visualized using a donut plot and waffle chart respectively. Google Data Commons was also queried using the python API for the minor allele frequency from dbGAP or the 1000 Genome Project when no dbGAP minor allele frequency was recorded. This data originally came from dbSNP and was visualized in a histogram.[62] The population specific minor allele frequency was obtained as part of output from SNiPA and were visualized in histograms using custom script.[54] The significant gene associations in whole blood, pancreas, and thyroid of an input list of genetic variants were also obtained by Google Data Commons using the python API. The original providence for this data is GTEx.

## Visualization of chromosomal location and GWAS significance

The chromosome location of the input and output genetic variant lists were visualized using a waffle chart generated from custom script. The proportion of output candidate genetic variants that were called significant by Type 1 Diabetes GWAS studies versus those that are in linkage disequilibrium with a significant variant were visualized in a donut plot using a custom script.

## Visualization of gene regulatory networks

Regulatory networks are also visualized by building bipartite graphs of regulatory element–gene pairs. The label of regulatory elements is the number of unique genetic variants contained within that regulatory element. The node color is determined by the type of regulatory element. The edge width is specified by the number of unique reads composing the chromatin contact call associating a regulatory element with the gene in close physical proximity. The visualization of each component in the graph is outputted. These provide insight into the structure and composition of regulatory networks in which genetic variants participate.

## Representation learning of the genetic variant–gene regulatory networks

The nodes of the bipartite graph of genetic variant–gene associations resulting from the SNP Prioritization Pipeline were converted to a matrix representation using node2vec.[63] Matrices of nodes belonging to the same component were combined to form a matrix that represented the components for all components composed ≥5 nodes. PCA was performed on the component matrices with n = 6 PCA components—only two of which were visualized (S6B Fig). The number of components was set as the minimum number of components needed to account for 80% of the variance in the data (n = 9). K-means clustering was performed on the PCA. K was optimized using the elbow plot method (k = 4). We named each cluster of gene regulatory networks based on the similarity in structure shared between the networks belonging to a given cluster.

## Selection of loci of interest

To validate the genetic variants, regulatory regions, and loci prioritized by the pipeline, we performed a deeper dive on a select number of loci (HLA, IL2RA, IKZF3, and IKZF1). HLA was selected due to having the most connected genetic variant and gene as well as being the most dense gene regulatory network (i.e. contained the most genetic variants and genes). It is also extensively studied thereby providing the opportunity to validate if the results of the pipeline were consistent with the literature. To further validate the pipeline we explored the IL2RA, IKZF3, and IKZF1 loci, which are all known to play a role in B-lymphocyte development and Type 1 Diabetes. These were additionally selected to provide examples of how to interpret the output of the pipeline at gene regulatory networks that had different structures.

## Visualization of chromatin connectivity

The chromatin connectivity at 5 kb resolution in a connected component of genetic variant–regulatory element–gene trios were visualized via a circus plot using a custom python script. Each node was a 5 kb section of the genome and the edges are distal connections between segments of the genome. The width of the edge correlates to the strength of the connectivity as represented by the number of unique reads.

The raw data of the chromatin connectivity at a given locus was visualized in virtual 4C plots using custom script.

## Supporting information

**S1 Fig. Schematic of data workflow.** Biomedical Data Commons was built by converting publicly available datasets into MCFs, which were then ingested into the knowledge graph. This makes the data searchable using the Data Commons API. Private multidimensional 'omics data was used as input into the SNP Prioritization Pipeline. This resulted in a list of candidate genetic variants, which were then used to generate queries to extract publicly available information on these variants from Biomedical Data Commons. Data and data processes relating to Biomedical Data Commons are in green and those related to the private data used in this study along with the developed SNP Prioritization Pipeline are in yellow. *denotes input data that is cell type-specific and needs to belong to the same cell type of interest.
(TIF)

**S2 Fig. SNP genetic variants are common, and their gene targets are associated with B-lymphocyte activity. A:** The code cyclomatic complexity score for the custom scripts used to identify the clinical significance, functional category, and significant gene association analyses using local data scientist approach involving data download (blue) or Data Commons (gold). **B:** Donut plot of genetic variants in which at least one gene target was verified by GTEx significant genetic variant—gene association in Whole Blood (burgandy), Thyroid (blue), Pancreas (gold) or a combination of tissues–Thyroid + Whole Blood (purple), Pancreas + Whole Blood (orange), or Pancreas + Thyroid + Whole Blood (turquoise). Genetic variants for which none of its gene targets were a GTEx significant genetic variant–gene association are in silver. **C:** Gene Ontology (GO) terms for genetic variants SNP pipeline identified gene targets. **D-H:** Minor allele frequency of pipeline genetic variants in specific subpopulations: African (**D**; green), Americans (**E**; blue), European (**F**; gold), East Asian (**G**; plum), and South Asian (**H**; turquoise). The red line is at minor allele frequency 0.02.
(TIF)

**S3 Fig. Workflow schematic of applying the SNP Prioritization Pipeline to B-lymphocyte data.** The original input list of genetic variants was generated by retrieving the Type 1 Diabetes significantly associated variants from GWAS Catalog (267) and then finding the genetic variants in linkage disequilibrium with those significant variants (12,707). Together these 12,974 variants were used as input into the SNP Prioritization Pipeline along with cell type-specific data on regulatory elements and 3D connectome as well as protein-coding genes genomic positions. The pipeline outputted associated genetic variants—regulatory elements—genes, which together formed trios. The number of unique genetic variants, regulatory elements, and genes that participate in these trio conformations in GM12878 cells using H3K27ac HiChIP input data are represented. The gene list from step 1 of the SNP Prioritization Pipeline is then used as input for Biomedical Data Commons queries. In addition, the trios generated in step 1 are used as input into step 2 of the SNP Prioritization Pipeline along with cell type-specific gene expression data. The output is visualizations of gene regulatory networks and a ranked

list of genetic variants and genes. Input and output data at each step of the pipeline is color coded by type of data: genetic variants (green), regulatory elements (blue), genes (gold), 3D connectome (black), gene expression data (magenta), gene regulatory networks (turquoise), and ranked list of genetic variants and genes (orange).
(TIF)

**S4 Fig. SNP Pipeline output with cohesin HiChIP input is a subset of the H3K27ac HiChIP output. A:** Venn diagram of trios generated with cohesin HiChIP (green) vs H3K27ac HiChIP (blue) with trios common to both datasets in green. **B:** Gene Ontology (GO) terms for genetic variants SNP pipeline identified gene targets with cohesin HiChIP as input. **C:** Kegg pathways for genetic variants SNP pipeline identified gene targets with cohesin HiChIP as input. **D:** Scatter plot of the top binding motifs of pipeline identified regulatory elements with cohesin HiChIP as input. **E:** Donut plot of the number of pipeline genetic variants that are significant in a Type 1 Diabetes GWAS study (green) or are in linkage disequilibrium with a significant genetic variant (silver) with cohesin HiChIP (left panel) or H3K27ac HiChIP (right panel) as input.
(TIF)

**S5 Fig. SNP Prioritization Pipeline produces similar results to the Sharp et al.** GRS2 study (2019). In addition to identical calls as the Sharp et al. GRS2 study, the SNP pipeline identifies variants (green) nearby GRS2 variants (blue) that have the same gene targets (2019). These variants in close physical proximity include those in IL27, IL2RA, UBASH3A<RASGRP1, CLEC16A, and RNLS loci. The difference between these two studies is that this connectome study (blue) restricts SNPs to regulatory elements and uses cell-type specific biological data as input whereas the GRS2 study (green) performs a statistical analysis of UK BioBank records to identify variants.
(TIF)

**S6 Fig. Four types of genetic variant–gene regulatory network component structures. A:** Heatmap of the closeness, degree, and betweenness centrality of the nodes in the genetic variant–gene regulatory networks. **B:** PCA of the matrix representation of the components of the genetic variant–gene regulatory network. K-means clustering (k = 4) was performed on the PCA and the boundaries are represented by the colors. Each black dot is a component and the white X's mark the centroid of a cluster. **C:** Components of the genetic variant (blue)–gene (gold) network that represent each cluster of components.
(TIF)

**S7 Fig. The HLA locus is the top candidate region identified by the SNP prioritization pipeline. A:** Zoomed in bipartite graph of the 1 Mbp HLA component containing top candidates HLA-DQB1, HLA-DRB1, and rs14004 with gene and genetic variants as nodes and chromatin connections as edges. Node color indicates closeness centrality score with gold being most connected and purple being least connected nodes in the graph. Gene nodes are labeled, and genetic variant nodes are unlabeled. **B:** Zoomed in bipartite graph of the 1 Mbp HLA component HLA component containing top candidates HLA-DQB1, HLA-DRB1, and rs14004 with gene and regulatory elements as nodes and chromatin connects as edges. Gene nodes are labeled and white. Regulatory element nodes are colored by type and labeled by the number of unique genetic variants contained in the regulatory element. The width of edges indicates connectivity strength as indicated by the number of unique HiChIP reads.
(TIF)

**S8 Fig. IKZF3 is a top pipeline identified candidate for T1D-associated genes. A:** Visualization of the IKZF3 component of interconnected pipeline genetic variant–regulatory element–gene trios (chr17: 39,635,000–40,835,000). B: Bipartite graph of the IKZF3 component with gene and genetic variants as nodes and chromatin connections as edges. Node color indicates closeness centrality score with gold being most connected and purple being least connected nodes in the graph. Gene nodes are labeled, and genetic variant nodes are unlabeled. **C:** Bipartite graph of IKZF3 component with gene and regulatory elements as nodes and chromatin connects as edges. Gene nodes are labeled and white. Regulatory element nodes are colored by type and labeled by the number of unique genetic variants contained in the regulatory element. The width of edges indicates connectivity strength as indicated by the number of unique HiChIP reads. **D:** Circos plot of the chromatin connectivity at 5 kb resolution in the IKZF3 locus. The nodes are sections of the genome and the edges are the chromatin connectivity with the width indicating connectivity strength. An asterisk labels the starting (chr17: 39,635,000; green) and terminating (chr17: 40,835,000; gold) nodes of the plot. GM12878 (left panel) and Treg (right panel) pipeline trio contacts are visualized.
(TIF)

**S9 Fig. IKZF1 identified as a candidate for T1D-associated genes by the pipeline. A:** Visualization of the IKZF1 component of interconnected pipeline genetic variant–regulatory element–gene trios (chr7: 50,300,000–50,545,000). B: Bipartite graph of the IKZF1 component with gene and genetic variants as nodes and chromatin connections as edges. Node color indicates closeness centrality score with gold being most connected and purple being least connected nodes in the graph. Gene nodes are labeled, and genetic variant nodes are unlabeled. **C:** Bipartite graph of IKZF1 component with gene and regulatory elements as nodes and chromatin connects as edges. Gene nodes are labeled and white. Regulatory element nodes are colored by type and labeled by the number of unique genetic variants contained in the regulatory element. The width of edges indicates connectivity strength as indicated by the number of unique HiChIP reads. **D:** Circos plot of the chromatin connectivity at 5 kb resolution in the IKZF1 locus. The nodes are sections of the genome and the edges are the chromatin connectivity with the width indicating connectivity strength. An asterisk labels the starting (chr7: 50,300,000; green) and terminating (chr7: 50,545,000; gold) nodes of the plot. GM12878 (left panel) and Treg (right panel) pipeline trio contacts are visualized.
(TIF)

**S1 Table. Main Biomedical Data Commons entity types.** The number of entities of each type and the databases contributing their underlying data. The number of entities in parentheses represent the number of additional nodes stored in a biomedical specific knowledge graph, but not the main knowledge graph. The databases contributing to each entity type are also reported.
(XLSX)

**S2 Table. Main Biomedical Data Commons edges.** Edges are defined as links between two nodes in the graph and the number edge present in the current graph is stated. Here edges are grouped into categories and the number of edges of each category are reported. The number of edges in parentheses represent the number of additional edges stored in a biomedical specific knowledge graph, but not the main knowledge graph. The number of types of nodes whose edges compose each edge category are also reported.
(XLSX)

**S3 Table. Ranked list of genetic variant and genes ordered by closeness centrality score calculated from their gene regulatory networks for GM12878 cells with H3K27ac HiChIP**

**connectome data.**
(XLSX)

**S4 Table. Genetic variants in close proximity to HLA-DQB1 or HLA-DRB1.** Genetic variants and their distance to the TSS of HLA-DQB1 and HLA-DRB1 if they are distally connected to either gene as identified by the SNP Pipeline. Their minor allele frequency according to the 1000 Genome Project and the GM12878 regulatory elements in which they reside.
(XLSX)

**S5 Table. ENCODE ChIP-seq datasets GEO accession numbers.** List of the ENCODE ChIP-seq datasets and their GEO accession numbers used in this study.
(XLSX)

**S6 Table. The entities along with their subclass and description that are uniquely defined by Biomedical Data Commons schema to represent the data in the knowledge graph.** This list was generated 4/9/21. For an up-to-date representation of the schema query the Biomedical Data Commons graph or browser or check the github repository.
(XLSX)

**S7 Table. The properties along with their domain, range, and description that are uniquely defined by Biomedical Data Commons schema to represent the data in the knowledge graph.** This list was generated 4/9/21. For an up-to-date representation of the schema query the Biomedical Data Commons graph or browser or check the github repository.
(XLSX)

**S8 Table. The enumeration subclass along with their description and their subclass types of plus associated descriptions that are uniquely defined by Biomedical Data Commons schema to represent the data in the knowledge graph.** This list was generated 4/9/21. For an up-to-date representation of the schema query the Biomedical Data Commons graph or browser or check the github repository.
(XLSX)

## Acknowledgments

We thank the Google Data Commons team for developing the underlying infrastructure used to build BMDC. We are also grateful for their help in developing the schema for BMDC. We acknowledge that we used publicly available information from the National Library of Medicine in this work.

## Author Contributions

**Conceptualization:** Samantha N. Piekos, Ramanathan V. Guha, Anthony E. Oro.

**Data curation:** Sadhana Gaddam.

**Formal analysis:** Samantha N. Piekos.

**Funding acquisition:** Ramanathan V. Guha, Anthony E. Oro.

**Software:** Samantha N. Piekos, Pranav Bhardwaj, Prashanth Radhakrishnan.

**Supervision:** Ramanathan V. Guha, Anthony E. Oro.

**Writing – original draft:** Samantha N. Piekos, Anthony E. Oro.

**Writing – review & editing:** Samantha N. Piekos, Prashanth Radhakrishnan, Anthony E. Oro.

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
