## [Decision Letter · Decision Letter 0]

28 Feb 2021

Dear Dr. Piekos,

Thank you very much for submitting your manuscript "Biomedical Data Commons (BMDC) prioritizes B-lymphocyte non-coding genetic variants in Type 1 Diabetes" for consideration at PLOS Computational Biology.

As with all papers reviewed by the journal, your manuscript was reviewed by members of the editorial board and by several independent reviewers. In light of the reviews (below this email), we would like to invite the resubmission of a significantly-revised version that takes into account the reviewers' comments.

We cannot make any decision about publication until we have seen the revised manuscript and your response to the reviewers' comments. Your revised manuscript is also likely to be sent to reviewers for further evaluation.

Sincerely,

Ilya Ioshikhes

Deputy Editor

PLOS Computational Biology

William Noble

Deputy Editor

PLOS Computational Biology

Reviewer's Responses to Questions

**Comments to the Authors:**

Reviewer #1: The authors have written a nice manuscript describing a new interface to understand clinically relevant high dimensional data and an example therein. The example is a vignette describing the use of orthogonal genetic and epigenetic datasets to elucidate the potential epigenetic consequences of common germ line SNPs in type I diabetes. They identify a majority occur in regulatory regions identified by ATAC-Seq and two HiCHIP datasets in B cells (most of which are confirmed in two T cell datasets). The data and methodology appear interesting. I would say that the manuscript itself is dense. It is difficult to follow because terminology is almost always the same even if the assays are different. I would highly encourage having supplementary figures for workflows and files for methods to describe how things were done. I describe a few examples below but there are more. I would encourage the authors to be proactive about this and not simply address these few examples i have listed here.

Major comments:

1. Though the authors show vignettes describing how there are some “B-cell specific” variants, i.e. those that occur in MHC class II loci, it is not clear if or why B-cell datasets are preferable to T cell datasets. IS the B-cell dataset preferable (meaning it explains a higher number of variants overall?)

2. The manuscript is too dense. It is difficult to follow what was exactly done. I’ll provide multiple examples here but the examples affect actually every aspect of the manuscript.

A) PCA plots in supplementary figure 4. What are each dots? They represent a node? How exactly were the nodes identified? PCA’s were generated in A and B seem similar except they use different legends? The legend for D looks similar to A and B but somehow succeeds C which is about centrality. The k-means clustering appears to be done after PCA, but the PC1 and PC2 change by this analysis? What was done here?

B) identifying the nodes. Presumably in supplementary figure 4E, the dense cluster plot indicates the reason why the authors focused on MHC class II. However, how exactly were IL2RA, IKZF3 chosen? The legends seems to indicate they were the top candidates chosen by different analyses but which analyses? Do they reflect different nodes in the cluster plots? How do these graphs connect to one another?

C) figure 2J. What are the output SNPs? The results describe utilizing SNPs in databases and then prioritizing/categorizing them based on their presence in various databases. I presume these are the input SNPs? What are the output SNPs? Are these the SNPs that are found by

D) Figure 1C. Genetic variant gene association. Presumably this is the regulatory region of the trio described in the text?

E) I can add more vignettes like this. I think the paper seems well done, but the writing is 1) so dense 2) leaves out salient details 3) uses almost identical terminology to describe different algorithms that lead to different results. I would highly recommend that the reviewers have someone who was not involved in the project read the paper and try to elucidate how the data was generated and represented. If the word limits preclude a sufficient description of the methods, i would include them in a supplementary document.

Minor comments:

1. Comments such as B-cell specific may be misleading. MHC class II is canonically expressed by professional antigen presenting cells such as B cells. T cells are not traditionally considered ones. However, it may be possible that this observation is not shared with T cells but may be shared with other cell types such as dendritic cells.

2. Scattered typos in the headers.

Reviewer #2: There are some minor points which need to be more expanded in the manuscript:

Methods:

Please explain the schema of your developed data base more, such as the relationship between the integrated databases.

In a part of your method please explain the data cleansing you done for the integration of data, how did you handle data type or data structure mismatch?

Please draw a comprehensive sketch of your developed ontology, it is very hard to follow the whole idea through the method part.

Discussion:

Please compare the advantages of you study in comparison to other studies.

What were the limitations of your study and how did you handle them?

Reviewer #3: This work aims at showcasing Biomedical Data Commons that here is used to query a knowledge graph of genomic, epigenomic, and transcriptomic data from seven databases. The field of application is type 1 diabetes – but the choice of analysis is on the less understood role of B lymphocytes. The basic concept is that of calling regulatory elements, mapping variants in a specific B cell line (GM12878) and assigning the regulated gene by H3K27ac HiChIP. There is then a number of statistical genetic assumptions and analyses that start with the assessment of variants that compose a genetic risk score of T1D based on GWAS. Individually, the various steps are conducted correctly.

There are a number of issues that diminish the impact of this work. They can be grouped in: biological rational, study design and statistical support, study interpretation.

1. Biological rational: there are several possibilities to benchmark BMDC: an area of advanced knowledge, or an unknown area of knowledge in a disease. T1D has a well-defined basis in T cell immunity and it would have been interesting and important to understand the meaning of the polygenic basis of disease in those cells. However, the authors chose the more complex path of B cell biology. The workhorse of the work is an immortalize B cell line that provides the data to support the analysis. The biological assumption is that this cell in some form, reflects the biology of a primary B cell in a patient. It is clear that that data currently available is what it is – but the decision to use immortalized B cells makes the assessment of result complex.

2. Study design and statistical support. It is really difficult to follow what has been done with variant selection. The manuscript refers to a list of 267 previously published significant T1D GWAS variants and an additional 12,707 variants in linkage disequilibrium. It also refers to the 67 variants in the genetic risk score. Then, there are 602 unique genetic variants participating in 8,682 unique genetic variant - regulatory element - gene trios. It is assumed that here, “variants” refer to the unique variants of the cell line and not to population variants. From this point on, there are a number of comments on prioritization that lacks a clear definition of the approach, the materials and tools to prioritize. There are a number of statements that more directly refer to overlaps/enrichment between population variants and lymphocyte variants – with unclear statistical statements. What is the basis for an statement of “strong connection”, etc.

There are also confusing statements such as:

Lines 219 to 226: “Genetic variants in non-coding regulatory elements - open chromatin regions or transcription factor binding sites - in GM12878 B-lymphocyte cells were associated with gene targets using three-dimensional chromatin conformation data defined by H3K27ac HiChIP……..The vast majority of these trio variants are of unknown clinical significance and they are enriched for non-coding regions compared to the input genetic variants.” - how can there be other than “enrichment” in non-coding regions: the input genetic variants are defined as being in the non-coding region.

Another confusing statement is:

“On top of this platform, we created a pipeline that prioritizes genetic variants, which are primarily of unknown clinical significance and in the non-coding genome. These variants are specific to B-lymphocytes but are associated with genes previously implicated in Type-1 Diabetes, suggesting they effect cell type-specific gene regulation.” – what is understood for variants specific to B-lymphocytes: variants are either specific to a host, or to a B cells line (via transformation or oncogenesis” – but there are no universal variants to B cells.

Down in the manuscript, there are many traditional pathway and clustering operations, assessment of GTEx etc, with a lack of clarity of what goes into what analysis and for which biological question. Of the 7 data sources, what was the exact use and contribution of each to the graph and the downstream analysis?

Perhaps the study would benefit from a clear figure that indicates what was done with what data and why.

3. Study interpretation. The text, and many of the very complex figures do not lend themselves to a clear interpretation. There are many variants and loci associated with T1D – at the end there are some proposed mappings to putative regulatory areas of the HLA-DR locus. The “pipeline identified the IL2RA gene regulatory network with IL2RA as the top candidate within the locus by closeness centrality”. These are known loci and it is not fully clear that they were identified in analysis that used prior knowledge, or in naïve analyses that would have mapped and prioritized these loci. It is possible that the authors have explained the exact approach in the manuscript – but the truth is that the reader may have a lot of difficulty to understand the steps of analysis and the underlying assumptions and decisions.

The study focus on B cells. However, comments such as the following creates doubts regarding the uniqueness of the B cell role given that many of the variants are described as shared with other cell lines (all in very generic terms): “Trio variants are largely cell-type specific to the input data used in the pipeline, but input data from GM12878, and 1° Naive T, Th17, and Treg cells mostly identified the same gene targets (Figure 2E)”

**Have all data underlying the figures and results presented in the manuscript been provided?**

Reviewer #1: Yes

Reviewer #2: Yes

Reviewer #3: Yes

PLOS authors have the option to publish the peer review history of their article (what does this mean?). If published, this will include your full peer review and any attached files.

Reviewer #1: No

Reviewer #2: **Yes: **Farkhondeh Asadi

Reviewer #3: No
---

## [Decision Letter · Decision Letter 1]

3 Jun 2021

Dear Dr. Piekos,

Thank you very much for submitting your manuscript "Biomedical Data Commons (BMDC) prioritizes B-lymphocyte non-coding genetic variants in Type 1 Diabetes" for consideration at PLOS Computational Biology. As with all papers reviewed by the journal, your manuscript was reviewed by members of the editorial board and by several independent reviewers. The reviewers appreciated the attention to an important topic. Based on the reviews, we are likely to accept this manuscript for publication, providing that you modify the manuscript according to the review recommendations.

Sincerely,

Ilya Ioshikhes

Deputy Editor

PLOS Computational Biology

William Noble

Deputy Editor

PLOS Computational Biology

[LINK]

Reviewer's Responses to Questions

**Comments to the Authors:**

Reviewer #1: The authors have adequately addressed our concerns. The manuscript is ready for publication.

Reviewer #2: no comments

Reviewer #3: Thanks for improving the readability of the manuscript.

One key comments, and a couple of minor remarks.

Two reviewers commented on the use of B cells instead of T cells. The authors provide an explanation in the anser to reviewers document that is perfectly acceptable: that this is the cell type that has extensive biological/genomic information. However, the paper provides a more novelty-based explanation implying that there is a biological interest. It would be preferable that the authors are as transparent with the readership and they were with the reviewers: the choice is a convenience decision, but may bring some biological insight on the role of B cells in diabetes, as well as indicating to what extent the work retrieved generalizable knowledge from B cells.

Minor:

1. "The vast majority of output candidate variants are not evolutionarily conserved as defined by CADD score". The CADD score includes conservation, but it does not define conservation - other metrics go into CADD

2. "Using the BMDC python API, we observed the majority of output candidate variants identified by the pipeline (S3 Supplementary Figure 3) to be common with a minor allele frequency (MAF) greater than 2%". By definition, if the input is GWAS information, it is expected that MAF will be 1-5%.

**Have the authors made all data and (if applicable) computational code underlying the findings in their manuscript fully available?**

Reviewer #1: Yes

Reviewer #2: Yes

Reviewer #3: Yes

PLOS authors have the option to publish the peer review history of their article (what does this mean?). If published, this will include your full peer review and any attached files.

Reviewer #1: No

Reviewer #2: **Yes: **Farkhondeh Asadi

Reviewer #3: **Yes: **Amalio Telenti

Figure Files:

Data Requirements:

Reproducibility:

References:

---

## [Decision Letter · Decision Letter 2]

25 Aug 2021

Dear Dr. Piekos,

We are pleased to inform you that your manuscript 'Biomedical Data Commons (BMDC) prioritizes B-lymphocyte non-coding genetic variants in Type 1 Diabetes' has been provisionally accepted for publication in PLOS Computational Biology.

Best regards,

Ilya Ioshikhes

Deputy Editor

PLOS Computational Biology

William Noble

Deputy Editor

PLOS Computational Biology

Reviewer's Responses to Questions

**Comments to the Authors:**

Reviewer #3: Thanks for the revisions.

**Have the authors made all data and (if applicable) computational code underlying the findings in their manuscript fully available?**

Reviewer #3: None

PLOS authors have the option to publish the peer review history of their article (what does this mean?). If published, this will include your full peer review and any attached files.

Reviewer #3: **Yes: **Amalio Telenti

---

## [Editor Report · Acceptance letter]

15 Sep 2021

PCOMPBIOL-D-20-02109R2 

Biomedical Data Commons (BMDC) prioritizes B-lymphocyte non-coding genetic variants in Type 1 Diabetes

Dear Dr Piekos,

I am pleased to inform you that your manuscript has been formally accepted for publication in PLOS Computational Biology. Your manuscript is now with our production department and you will be notified of the publication date in due course.

With kind regards,

Olena Szabo
